# Phytochemicals from *Pterocarpus angolensis* DC and Their Cytotoxic Activities against Breast Cancer Cells

**DOI:** 10.3390/plants13020301

**Published:** 2024-01-19

**Authors:** Zecarias W. Teclegeorgish, Ntebogeng S. Mokgalaka, Douglas Kemboi, Rui W. M. Krause, Xavier Siwe-Noundou, Getrude R. Nyemba, Candace Davison, Jo-Anne de la Mare, Vuyelwa J. Tembu

**Affiliations:** 1Department of Chemistry, Tshwane University of Technology, Private Bag X680, Pretoria 0083, South Africa; zakighedil@gmail.com (Z.W.T.); mokgalakans@tut.ac.za (N.S.M.); 2Department of Chemistry, University of Kabianga, Kericho 2030-20200, Kenya; 3Department of Chemistry, Rhodes University, Makhanda 6140, South Africa; r.krause@ru.ac.za; 4Department of Pharmaceutical Science, Sefako Makgatho Health Science University, P.O. Box 60, Medunsa, Pretoria 0204, South Africa; xavier.siwenoundou@smu.ac.za; 5Department of Biochemistry and Microbiology, Female Cancers Research at Rhodes University (FemCR2U), Makhanda 6140, South Africa; g21n4142@campus.ru.ac.za (G.R.N.); c.davison@ru.ac.za (C.D.)

**Keywords:** *Pterocarpus anglonesis*, cytotoxicity, HCC70, MCF-7, MCF-12A, breast cancer

## Abstract

*Pterocarpus anglonesis* DC is an indigenous medicinal plant belonging to the *Pterocarpus* genus of the Fabaceae family. It is used to treat stomach problems, headaches, mouth ulcers, malaria, blackwater fever, gonorrhea, ringworm, diarrhea, heavy menstruation, and breast milk stimulation. Column chromatography of the stem bark extracts resulted in the isolation of eight compounds, which included friedelan-3-one (**1**), 3α-hydroxyfriedel-2-one (**2**), 3-hydroxyfriedel-3-en-2-one (**3**), lup-20(29)-en-3-ol (**4**), Stigmasta-5-22-dien-3-ol (**5**), 4-*O*-methylangolensis (**6**), (3*β*)-3-acetoxyolean-12-en-28-oic acid (**7**), and tetradecyl (E)-ferulate (**8**). The structures were established based on NMR, IR, and MS spectroscopic analyses. Triple-negative breast cancer (HCC70), hormone receptor-positive breast cancer (MCF-7), and non-cancerous mammary epithelial cell lines (MCF-12A) were used to test the compounds’ cytotoxicity. Overall, the compounds showed either no toxicity or very low toxicity to all three cell lines tested, except for the moderate toxicity displayed by lupeol (**4**) towards the non-cancerous MCF-12A cells, with an IC_50_ value of 36.60 μM. Compound (3*β*)-3-acetoxyolean-12-en-28-oic acid (**7**) was more toxic towards hormone-responsive (MCF-7) breast cancer cells than either triple-negative breast cancer (HCC70) or non-cancerous breast epithelial (MCF-12A) cells (IC_50_ values of 83.06 vs. 146.80 and 143.00 μM, respectively).

## 1. Introduction

*Pterocarpus* of the *Fabaceae* (Leguminosae) family comprises 41 species mostly of timber trees found in the tropical parts of the world and native to Africa, South America, and Asia [1]. The *Pterocarpus* genus is known to produce secondary metabolites, including flavonoids, isoflavonoids, pterocarpans, auriones, lignans, stilbenes, sterols, sesquiterpenes, and triterpenes [2]. *Pterocarpus angolensis* DC (commonly known as Kiaat in Afrikaans; African teak or wild teak, in English; Morôtô, in seSotho; Mokwa in SeTwana, Mutondo, in TshiVenda; Mukwa or Mubvamaropa in Shona and Umvangazi in isiZulu) grows in the warm, semi-arid areas in the northeast of South Africa, Mozambique, Zimbabwe, northern Botswana, and Namibia and other northern parts of Africa [3]. It grows in subtropical woodlands where the rainfall is above 500 mm per annum [3]. The tree can be found in eastern and southern Africa at elevations ranging from sea level to roughly 1650 m in all types of woodland and wooded savannah [4]. *P. angolensis* is used for a variety of purposes in folk medicine. A cold infusion is used to treat stomach problems, headaches, and mouth ulcers [5]. Its decoction of bark and infusion of roots are often used in the treatment of different ailments such as to cure malaria, blackwater fever, gonorrhea, heavy menstruation, blood in the urine, and schistosomiasis [5,6]. The tree is a superb wood that is resistant to termites, marine borers, and beetles [7]. This wood is mostly used for furniture, joinery, and decorative veneer [3,5].

Phytochemical constituents from *P. angolensis* DC were previously isolated and identified as isoflavonoids, epicatechins, dexoybenzoin, sterols, triterpenes, chalcones, and fatty acids [8,9]. For instance, flavonoids, including muningin [8], Prunetin [10], 4-*O*-methyl angolensin [11], hexamer of *epi*catechin, (-)-*epi*catechin, *epi*catechin (4-*β*-8)-*epi*catechin (B2), and an *epi*catechin-3-*O*-galate [9], were isolated from bark *P. angolensis*. Other compounds, including n-hexadecanoic acid, hexadecanoic acid methyl ester, octadecanoic acid, 7-dehydrodiosgenin, stigmasta-3,5-dien-7-one, lup-20(29)-en-3ol, and friedelan-3-one, were identified from the hexane extract of the stem bark *P. angolensis* [12].

The compounds from *P. angolensis* exhibited antibacterial properties with the minimum inhibition concentration (MIC) of epicatechin-3-*O*-gallate 50 µg/mL against *Staphylococcus aureus* and *Acinetobacter calcaoceuticus* and *hexamer of epicatechin 25* µg/mL against *Staphylococcus typhi* and *Kocuria kristinae* [9]. Additionally, the minimum bactericidal concentration (MBC) and minimum fungicidal concentration (MFC) values of methanol and dichloromethane (DCM) extract against bacteria (*S. aureus* and *S. aglaactiae*) and fungus (*Candida krusei*) varied from 0.166 g/mL to 0.0417 g/mL, respectively [13]. The (DCM/Methanol *v*/*v*, 50%) extract of *P. angolensis* inhibited *Plasmodium falciparum* chaperones (PfHsp70-z and PfHsp70-1), with IC_50_ values of 13.87 and 0.20 µg/mL, respectively [14]. The crude aqueous extracts of bark and root from *P. angolensis* enabled the healing of breached tissue and induced an increase in chondrogenesis [15]. According to our knowledge, no anticancer activities have been reported on the extracts and isolated compounds of *P. angolensis*. Hence, this study focused on the cytotoxicity properties of the crude extract and its phytochemicals. Furthermore, there is no report on the geographical variation in metabolites present in *P. angolensis* ecotypes.

## 2. Results and Discussion

### 2.1. Isolated Compounds

The phytochemical investigations of *P. angolensis* stem bark extract allowed for the isolation of eight compounds, which are six triterpenes, including friedelan-3-one (**1**), 3-Hydroxyfriedel-2-one (**2**), 3α-Hydroxyfriedel-3-en-2one (**3**), (3β)-lup-20(29)-en-3-ol (**4**), stigmasta-5-22-dien-3-ol (**5**), and (3β)-3-acetoxyolean-12-en-28-oic acid (**7**), one of which is a deoxybenzoin known as (±)-4-*O*-methylangolensin (**6**) and the other an alkyl hydrocinnamte group known as tetradecyl (E)-ferulate (**8**). The chemical structures of the eight obtained compounds from the wild and cultivated stem bark of *P. angolensis* are illustrated in Figure 1.

Compounds **1**, **2**, **3**, **4**, **5**, **6**, and **8** were isolated from the DCM crude extract of cultivated *P. angolensis*, while five compounds, **1**, **2**, **4**, **6**, and **8**, were obtained from the wild crude DCM extract. Compounds **6** and **8** were also isolated from hexane crude extracts, both cultivated and wild, respectively. Compounds **5**, **6**, and **7** were also isolated from crude ethyl acetate extracts from both wild and cultivated sources. This variation in the number of isolated compounds from the cultivated plants (**7**) compared to the wild plants (**5**) could be attributed to the differences in the geological location and climate conditions of the two populations. Limpopo Province, where the wild populations were collected, is much warmer than the Gauteng Province, where cultivated specimens were collected. Geographical location is known to influence the secondary metabolites profile of plants [16].

### 2.2. Physical Properties of the Isolated Compounds

*Friedelan-3-one* (**1**) is a white crystalline solid needle with a yield of 1.6687 g (0.035% m/m), Rf = 0.36 at 70/30% DCM/Hex, UV_254_-active dark spots in TLC.

*3α-Hydroxyfriedel-2-one* (**2**) is a white powder. The yield was 784.7 mg (0.0016%, m/m), UV_254-_ active, Rf = 0.54 100% DCM, purple spots on TLC.

*3-Hydroxyfriedel-3-en-2-one* (**3**) is a white crystalline needle-like solid that yielded 120.62 mg (0.005%, m/m), UV_254_-active, Rf = 0.27 at 100% DCM, yellow spot at TLC.

*(3β)-lup-20(29)-en-3-ol* (**4**) is a white powder with a yield of 51.30 mg (0.0021% m/m) and UV_254_ active, Rf = 0.45 at Hexane/DCM 20/80%, yellow spot on TLC.

*Stigmasta-5-22-dien-3-ol* (**5**) is a white amorphous solid that yielded 83.70 mg (0.0017%, m/m); Rf = 0.76 at 100% DCM, dark spot on TLC.

(±)-4-*O*-*Methylangolensin* (**6**) is a yellowish-orange solid with a mass of 175.20 mg (0.0037% m/m) and UV_254_-active Rf = 0.75 at Hex/DCM 10/90%, orange-yellowish spot on TLC.

(3β)-3*-Acetoxyolean-12-en-28-oic acid* (**7**) is a white powder. The yield was 19.24 mg (0.0012%, m/m) and is UV_254_- active, Rf = 0.75 Hex/DCM at 10/90%, yellow spot on TLC.

*Tetradecyl (E)-ferulate* (**8**) is a white-yellowish powder with a yield of 27.94 mg (0.0012% m/m), UV_254_-active, Rf = 0.39 at 100% DCM, dark spot on TLC.

The physiochemical properties of compounds **1**–**8**, including the IR, optical rotation, melting points, and molecular mass, are presented in Table 1 below.

### 2.3. Structural Analysis

*Friedelan-3-one* (**1**) was isolated as a white crystalline solid needle with pseudo-molecular ion [M + H]^+^ at *m/z* 427.3805, which indicated the molecular formula of C_30_H_50_O. The ^1^H NMR spectrum (Appendix A) of compound 1 showed seven methyl singlet proton resonances and one methyl doublet proton resonance at δ 1.17 (3H, s), 1.04 (3H, s), 1.00 (3H, s), 0.99 (3H, s), 0.94 (3H, s), 0.86 (3H, s), 0.86 (3H, s), and 0.71 (3H, s) ppm, respectively. One methyl resonance doublet at *δ* 0.86 (3H, s) ppm, *J =* 6.0 Hz was seen. The ^13^C NMR and DEPT spectra revealed the presence of 30 carbon resonances attributed to eight methyls, ten methylene, three methine, and nine quaternary carbon atoms, as indicated in Appendix A. The downfield carbon resonance at δ 213.2 ppm is a typical ketone group complemented by the carbonyl stretch of ketone (-C=O) present in the IR spectrum displaying at 1715.2 cm^−1^ assigned to C-3. The spectral data suggested compound **1** to be Friedelan-3-one, which was confirmed by the comparison of ^1^H NMR and ^13^C NMR with the literature. The compound (**1**) was previously isolated from the stem bark of the *Peritassa rompata* plant [17].

*3α-hydroxyfriedel-2-one* (**2**) was isolated as a white powder and a pseudo molecular ion [M + H]^+^ peak at *m/z* 443.3447 corresponding to the molecular formula of C_30_H_50_O_2_. The ^1^H NMR spectrum showed seven methyl proton singlet resonances at δ 0.87 (3H, s), 0.94 (3H, s), 0.97 (3H, s), 0.98, 1.00 (3H, s), 1.05 (3H, s), and 1.17 (3H, s) ppm and a doublet methyl proton resonance at δ 1.03 (3H, s) ppm. The ^1^H NMR spectrum also revealed a doublet oxy-methine (1H, dd *J* = 11.8, 2.98 Hz) proton resonance at δ 3.81 ppm (H-3). The ^13^C NMR spectrum showed eight methyls, ten methylene, three methine, and nine quaternary carbon atoms, as indicated in Appendix A. A downfield carbon resonance was observed in the carbon spectrum at δ 211.9 ppm; this supported the presence of a sharp absorption band observed at 1662.6 cm^−1^ in the IR spectrum. Another difference observed in compound **2** was the hydroxyl (OH) and its electronegative effect due to the carbonyl stretch of the ketone (at 1662.6 cm^−1^) that is lower than the typical ketone, which appears at 1715 cm^−1^ as detected in compound **1** [18]. Based on the NMR spectral data as well as the comparison with the literature [19], compound **2** was identified as 3-hydroxyfriedel-2-one.

*3-Hydroxyfriedel-3-en-2one* (**3**) was isolated as a white crystalline needle-like solid with a pseudo molecular ion [M + H]^+^ at *m*/*z* 441.2928 that corresponds to a molecular formula of C_30_H_48_O_2_ (calculated for *m*/*z* 440.3651). 3-hydroxyfriedel-3-en-2one (**3**) is a striking derivative of compound **2** (3-hydroxyfriedel-2-one), and the ^13^C NMR of compound **3** showed downfield *sp3* quaternary carbon resonance of an alkene at δ 142.6 ppm and δ 140.8 ppm, which were assigned to C-3 and C-4, respectively. This assignment was in agreement with the IR spectrum displaying a sharp band C=C vibration at 1466.3 cm^−1^. The double bond is missing in compound **2** because the alkene resonance is absent in the carbon spectrum. More differences were depicted: the chemical shifts in ring A of compound **3**. The presence of H-3 in compound **2** previously assigned at δ 3.81 (1H, dd *J* = 11.8, 2.98 Hz), and H-4 at δ 1.29 ppm were also missing in compound 3 in the ^1^H-NMR spectrum (Appendix A). Using spectral data, compound **3**, also identified as 3-hydroxyfriedel-3-en-2-one, was previously isolated from the stem bark of *Mallotus philippensis* [20]. The compound (**3**) was, however, isolated for the first time from *P. angolensis* in this study, which was supported by a comparison of spectroscopic data with values from the literature [19,21].

*(3β)-lup-20(29)-en-3-ol* (**4**) was isolated as a white powder and exhibited a molecular ion [M + H]^+^ peak at *m*/*z* 427.3677 with a double bond equivalent of six and the molecular formula of C_30_H_50_O (calculated for *m*/*z* 426.3861). The ^1^H NMR spectrum revealed an oxymethine proton resonance at δ 3.159 (H-3) ppm (dd, *J* = 4.4; 6.4 Hz) and seven methyl singlets resonance at δ 0.96 (3H, s), 0.76 (3H, s), 0.83 (3H, s), 1.03 (3H, s), 0.94 (3H, s), 0.78 (3H, s), and 1.67 (3H, s) ppm (Appendix A). ^13^C NMR/DEPT spectrum revealed 30 carbon resonances, which were shown to be seven methyl, eleven methylene, six methine carbons, and six quaternary carbons (Appendix A). The ^1^H NMR spectrum showed resonances at δ 4.68 ppm and δ 4.55 ppm attributed to the olefinic methylene protons assigned to H-29A and H-29B, indicating the presence of an isoprene unit; this is characteristic of the lupane triterpenoid class. The ^13^C NMR spectrum revealed that 79.02 ppm is typical of carbon attached to oxygen, δ 109.3 ppm (C-20) for methyl carbon resonance, and δ 151.0 ppm (C-29) for quaternary carbon resonance. The spectra evidence of compound 4 were comparable to the literature data for lupeol isolated previously from the seeds of *Heritiera utilis* [22] and *Amsonia grandiflora* [23]. Hence, the structure of compound 4 is confirmed to be a triterpenoid known as 3β-Lup-20(29)-en-3-ol (lupeol).

*Stigmasta-5-22-dien-3-ol* (**5**) was isolated as a white crystalline solid with a molecular ion [M + H]^+^ peak at *m/z* 413.3239, which corresponded to the molecular formula C_29_H_48_O with a double bond equivalent of six. The ^13^C NMR spectrum revealed twenty-nine carbon resonances, including eleven methine, nine methylene, six methyl, and three quaternary carbon resonances (Appendix A). The ^1^HNMR spectrum revealed six methyl proton resonances: δ 0.69 (3H, s), δ 0.80 (3H, s), δ 0.80 (3H, s), δ 1.02 (3H, s), δ 0.83 (3H, s), and δ 1.03 (3H, s) ppm. Furthermore, three olefinic proton resonance at δ 5.34 (d, *J* = 5.2 Hz, H-6), δ 5.14 (1H, dd, *J* = 8.5, 6.6 Hz. H-22), and δ 5.05 (1H, dd, *J* = 8.5, 6.4 Hz, H-23) and a triplet of a doublet of doublet at δ 3.51 (*J* = 6.1, 4.4, 5.1 Hz, H-3) were observed in the ^1^HNMR spectrum. Based on the spectral data and comparison with the values from the literature, compound **5** was determined to be a phytosterol known as stigmasterol, which is found in plant fats or oils, several therapeutic plants, and many vegetables [24].

(±)-*4*-*O*-*methylangolensin* (**6**) was obtained as a yellowish-orange solid and a pseudo molecular ion [M + H]^+^ at *m/z* 287.1341, which indicated the compound’s molecular formula is C_17_H_18_O_4,_ with a double bond equivalent of nine. The ^13^C NMR showed seventeen carbon resonances, including two aromatic rings, two methoxy’s, and a propyl ether, as shown in Appendix A. The ^1^H NMR (Appendix A) showed downfield a phenolic OH group resonance at δ 13.05 (1H, s, OH-2), two methoxy group resonances at δ 3.76 ppm (3H, s, 4-OCH_3_ 4′-OCH_3_), one methyl singlet at δ 1.50 ppm (3H, s, H-9), and methine proton resonances at δ 7.70 ppm (1H, d, *J* = 8.0 Hz, H-6), δ 6.36 (1H, d, *J* = 2.36 Hz, H-3), δ 6.33 (1H, dd, *J* = 8.0, 4.0 Hz, H-5), δ 4.60 (1H, q, *J =* 8.0 Hz, H-8), δ 7.21 (1H, d, *J =* 7.0 Hz, H-2′, H-6′), and δ 6.85 (1H, d, *J =* 6.96 Hz, H-3, H-5′). The ^13^C NMR showed two aromatic carbon resonances for rings A and B as 132.7 ppm (C-6), 162.4 ppm (C-4), and 165.9 ppm (C-2) and 128.6 ppm (C-2′, C-6′), 158.7 ppm (C-4′), and 114.4 ppm (C3′, C5′), respectively. The downfield carbonyl group resonance observed at δ 205.0 ppm (C-7) is a typical ketone group. Compound 6 was characterized as a 4-*O*-methylangolensis that belongs to a deoxybenzoin based on physical and spectral evidence [25,26].

*(3β)-3-Acetoxyolean-12-en-28-oic acid* (**7**) was isolated as a white powder and a pseudo molecular ion [M + H]^+^ at *m*/*z* 499.3795, which revealed the compound has a molecular formula C_32_H_50_O_4_, with a double bond equivalent of eight. The ^1^H NMR spectrum revealed seven methyl resonance, δ 1.13 (3H, s), 0.94 (3H, s), 0.93 (3H, s), 0.90 (3H, s), 0.86 (3H, s), 0.85 (3H, s) and 0.75 (3H, s) ppm, and an olefinic proton resonance, δ 5.28 ppm (1H, t, *J* = 3.7Hz, H-12), which is a characteristic of an olean-12ene skeleton. The ^13^C NMR spectrum revealed that the olefinic carbon resonances of δ 122.6 ppm (C-12) and δ 143.6 ppm (C-13) were assigned to C-12 and C-13, respectively. The ^13^C NMR revealed a resonance of δ 183.5 ppm (C-28), indicating the presence of a carboxylic group, and the resonance of δ 171.0 ppm (C-2′), a typical acetate. ^1^H NMR spectrum displayed a downfield proton resonance at δ 4.51 ppm (1H, t, *J* = 7.6 Hz, H-3) and a methyl proton resonance at δ2.04 ppm (3H, s, H-1′) due to C-3, which is consistent with an ester group, accounting for the additional two carbon in structure. This is further complemented in the IR spectrum with a sharp absorption band at 1732 cm^−1^ [16]. The ^13^C NMR spectrum showed that the thirty-two carbon resonances included seven methyl carbon, ten methylene carbon, five methine carbon, and nine quaternary carbons, as shown in Appendix A. The spectroscopic analysis and comparative literature study confirmed that compound **7** was identified as 3β-3-acetoxyolean-12-en-28-oic acid [27]. 3β-3-acetoxyolean-12-en-28-oic acid was isolated previously from the bark of *Quercus crispula*, commonly known as Mongolian oak [27].

*Tetradecyl (E)-ferulate* (**8**), was isolated as a white powder and showed a molecular ion [M + H]^+^ peak at *m/z* 391.2914 that corresponds to the molecular formula of C_24_H_38_O_4_. The ^1^HNMR spectrum (Appendix A) showed a saturated chain of methylene group at resonance at δ 1.25–1.27 ppm (2H, m, 1′-13′), terminal methyl proton resonance at δ 0.88 (3H, t, *J* = 6.6 Hz, 14′), three aromatic resonances at δ 7.07 ppm (1H, dd, *J* = 8.2, 1.6 Hz, H-2), δ 6.92 ppm (1H, d, *J* = 8.1 Hz, H-3), and δ 7.03 ppm (1H, d, *J* = 2.0 Hz, H-6), and one methoxy singlet resonance at δ 3.92 ppm (3H, s, H-15′). The two alkyl olefinic proton resonances of δ 7.63 ppm (1H, d, *J* = 15.9 Hz, H-7) and δ 6.31 ppm (1H, d, *J* = 15.9 Hz, H-8) were seen coupling in *trans* to each other, which is supported by the coupling constant value of *J* = 15.9 Hz. The ^13^C NMR showed twenty carbon resonances, including six phenyl groups, two olefinic, one methoxy, one methyl, and thirteen aliphatic chain carbon, as shown in Appendix A. The ^1^H NMR and ^13^C NMR resonances showed a typical structure of a ferulate ester attached to the alkenyl side chain of the compound. Compound **8** was thus identified as tetradecyl (*E*)-ferulate and also known as tetradecyl (*E*)-3-(4-hydroxy-3-methoxy-phenyl) prop-2-enoate [28]. Tetradecyl (*E*)-ferulate was previously isolated from the root of *Jatropha gosspifolia* [29].

### 2.4. Cytotoxicity Activity

The cytotoxicity of the pure compounds was evaluated using the in vitro cell culture against triple-negative breast cancer (HCC70) cell lines, hormone receptor-positive breast cancer (MCF-7), and non-tumorigenic epithelial cell lines derived from breast tissue (MCF-12A). A compound’s ability to inhibit a particular biochemical function is determined by its half-maximum inhibitory concentration (IC_50_). Etoposide was used as the positive control. The IC_50_ for each compound against the various cell lines under investigation is listed in Table 2.

Friedelan-3-one (**1**) was found to be non-toxic against breast cancer cell lines (MCF-7) and triple-negative cancer cell lines (HCC70) at the highest concentration of 200 µM and showed moderate cytotoxicity against non-tumorigenic epithelial cell lines (MCF-12A) at a concentration of 43.86 µM (Table 1). Previous studies, however, reported that friedelan-3-one (**1**) showed good activity against MCF-7 (1.2 µM) at 48 h [30] and exhibited the inhibition of CYP3A4 (10.79 µM) and 2E1 human liver microsomal cancer cells (22.54 µM) [31].

3α-Hydroxyfriedel-2-one (**2**) and (3*β*)-lup-20(29)-en-3-ol (**4**) were found non-toxic against HCC70 and showed low cytotoxicity against MCF-7 with IC_50_ values of 84.20 µM/mol and 148.7 µM/mol, respectively. 3α-Hydroxyfriedel-2-one (**2**) showed a strong cytotoxicity effect on the insect-derived sf9 (8.32 µg/mL) and mammalian CHO cells (50 > EC_50_ > 25 µg/mL) and no phytotoxicity against germination of *Lactuca sativa* seeds [32]. (3*β*)-lup-20(29)-en-3-ol (**4**) previously exhibited cytotoxicity with IC_50_ at 13.60–69.39 µM against glioblastoma U87MG.EGFR cells, CEM/ADR5000 leukemia cells, and IC_50_ in the range of 0.02-66.83 µM for CCRF-CEM cells [33].

3-hydroxyfriedel-3en-2-one (**3**) possessed low cytotoxicity activity (IC_50_) against HCC70 cell lines (146.4 µM) and 136.0 µM and moderate activity against (100.8 µM), which also showed a relatively good selectivity index = 1.35 (Table 1). The previous study of 3-Hydroxyfriedel-3en-2-one (**3**) showed no cytotoxicity (IC_50_ = 330 mol) for the inhibitory effects on the Epstein–Barr virus [34].

Stigmasta-5-22-dien-3-ol (**5**) showed a low cytotoxicity effect against HCC70 (IC_50_ = 109.4 µM) and a moderate effect against MCF-12A (IC_50_ = 44.14 µM). The isolated stigmasta-5-22-dien-3-ol (**5**) and tetradecyl (*E*)-ferulate (**8**), with IC_50_ values of 226.9 µM and 223.1 µM, respectively, did not show any cytotoxicity activity against MCF-7 (Table 1). In another study, Stigmasta-5-22-dien-3-ol (**5**) showed cytotoxicity effects against oral epithelial carcinoma cells (KB/C152) with an IC_50_ value of 81.18 μg/mL [20] and against HeLa cervical cancer cells with an IC_50_ value of 26.42 µM [35]. (3*β*)-3-Acetoxyolean-12-en-28-oic acid (**7**) and tetradecyl (*E*)-ferulate (**8**) exhibited low cytotoxicity against triple-negative breast cancer (HCC70) with IC_50_ values of 146.8 µM and 128.2 µM, respectively.

4-*O*-Methyl angolensin (**6**) exhibited a good selectivity index (SI = 1.32) but had low cytotoxicity (IC_50_ = 111.4 µM) against triple-negative breast cancer cells, HCC70. However, the compounds showed good cytotoxicity against breast cancer cells, MCF-7 with an IC_50_ value of 77,46 µM and a relatively high selectivity index (SI = 1.91) (Table 2). To the best of our knowledge, no studies have examined the anticancer cytotoxicity of 4-*O*-methyl angolensin (**6**). (3β)-3-acetoxyolean-12-en-28-oic acid (**7**) showed moderate cytotoxicity against MFC-7 cell lines with IC_50_ value of 83.06 µM and low activity against MFC-12A with an IC_50_ of 143.0 µM. In another study, (3β)-3-acetoxyolean-12-en-28-oic acid (**7**) exhibited cytotoxicity toward a human ovarian cancer cell line, SKOV3, and a human endometrial carcinoma cell line HEC-1A cells with IC_50_ values of 8.3 and 0.8 μM, respectively [36].

## 3. Materials and Methods

### 3.1. General Experimental Procedure

Column chromatography was performed on polyamide columns (Germany GmbH) over silica gel (Kieselgel 60 GF254, pore size 35–75 m particle size, Merck, Darmstadt, Germany), while thin-layer chromatography (TLC) was performed on Kieselgel 60 F254 (Merck) to a thickness of 0.25 mm. Under ultraviolet (UV) light, active spots on UV active silica gel were visualized (245 and 336 nm). All reagents and solvents (hexane, ethyl acetate, dichloromethane, and absolute ethanol) used in column chromatography were purchased from Merck and Sigma in South Africa and used exactly as received.

### 3.2. Plant Material

The natural (wild) habitat of *P. angolensis* in South Africa was spotted on the map and data of the South African Natural Biodiversity Institute (SANBI). The stem bark of mature *Pterocarpus angolensis* was collected from Nkowankowa (23.826923° S, 30.313919° E) in Limpopo province with an average temperature of 29 °C at the time of sampling. The plant was sampled from five different locations within a 500 m range of the area along the R71 roadside 55 km away from the town of Tzaneen towards Phalaborwa. The stem bark of mature cultivated *P. angolensis* was also collected from the Pretoria National Botanical Gardens (25.7395° S, 28.2733° E) in Gauteng province with an average temperature of 26 °C at the time of sampling. A voucher specimen of the plant (leaves, bark, and seeds) for identification with allocated number 18077 was deposited in the South African Natural Biodiversity Institute (SANBI).

### 3.3. Sequential Extraction

*Pterocarpus angolensis* plant material (stem bark of wild and cultivated trees) was cleaned with deionized water after being washed with tap water to remove dust. The stem bark was then cut into small pieces and placed in a laboratory room for two weeks at room temperature. The dried stem bark of the plant was ground into small pieces using a Retsch mill (Retsch SM 100, Haan, Germany). The small pieces of stem bark of *P. angolensis* were ground into a fine powder using a 1000 W Polymix PX-MFC 90D mill (Kinematica, Malters, Switzerland). The fine powdered plant material of 2.4 kg was weighed and added to 10 L round bottom flasks with 7500 mL organic solvents for extraction. The plant material was extracted sequentially with four analytical-grade organic solvents with increasing polarity order: Hexane (Hex) < dichloromethane (DCM) < ethyl acetate (EtOAc) < methanol (MeOH). The mixtures were shaken at room temperature for 48 h on an HS 501 horizontal shaker IKA (HS 501 digital IKA Werke, Staufen, Germany). A Whatman’s No. 1 filter paper was used to filter the extracts. This step was carried out in duplicate for maximum extraction. All filtrates of extracts were concentrated using a G1 diagonal glassware Rotary Evaporator (Heidolph Hei-VAP, Heidelberg, Germany) set to a constant temperature of 37 °C. In a fume hood, the concentrated crude extracts of wild and cultivated *P. angolensis* were dried, the wild plant yielded 17.48 g (Hex), 39.02 g (DCM), 19.21 g (EtOAc), and 388.30 g (MeOH), whereas the cultivated plant yielded 21.82 g (Hex), 42.45 g (DCM), 15.35 g (EtOAc), and 375.36 g (MeOH). The dried extracts of *P. angolensis* were stored at −10 °C to prevent any decomposition.

### 3.4. Nuclear Magnetic Resonance Spectroscopy

Moreover, 5.0–10.0 mg sample of each pure isolated compound of *P. angolensis* was dissolved in deuterated chloroform, CDCI_3,_ and transferred into an NMR tube with a diameter of 5 mm, a precision of medium wall, a frequency limit of 400 MHz, and a length of 7 inches (Wilmad^®^, Vineland, NJ, USA). Then, 1D and 2D NMR were acquired with gradient enhancement using a Varian 400 MHz premium-shielded NMR spectrometer (400/54/ASP, Santa Clara, CA, USA). Chemical shifts were measured relative to 7.2 ppm (^1^H) and 77.0 ppm (^13^C) NMR to deuterated chloroform, and structures of isolated compounds were proposed based on spectra interpretation [18].

### 3.5. Infrared Spectroscopy

Approximately 1.0 mg of isolated pure compounds of *P. angolensis* were analyzed using a Spectrum Two Universal (FTIR) spectrophotometer (Perkin Elmer, Waltham, MA, USA) to determine the functional groups present. For each sample, 32 scans in the mid-IR band of 4000–550 cm^−1^ were performed, and a background spectrum was obtained before each sample spectrum.

### 3.6. Mass Spectroscopy

High-resolution LC-MS spectra were collected using a Bruker Daltonics Compact QTOF Mass Spectrometer (Billerica, MA, USA) with an electrospray ionization probe in the positive mode (ESI+). Thermo Scientific Ultimate 3000 Dionex UHPLC (Milford, CT, USA) system with an Acclaim RSLC 120, C18, 2.2 m, 2.1 100 mm (P/N 068982) column was used for separation. The system’s components included the Detector DAD-3000 RS, Pump HPG-3400 RS, and Auto Sampler WPS-3000 RS (Milford, CT, USA) The solvent mixture was water–acetonitrile (10:90, *v*/*v*), with each solvent containing 0.1% of formic acid. Isocratic elution with a 5 min run time and injection volume of 3 μL was used.

### 3.7. In Vitro Cytotoxicity Assay

#### The Resazurin Assay

The cell viability after treatment with the isolated pure compounds of *P. angolensis* was analyzed using the resazurin assay in commercially available established cell lines purchased from the American Type Culture Collection [37]. Triple-negative breast cancer cell line (HCC70) (estrogen receptor (ER)^−^, progesterone receptor (PR), (HER-2)^−^, (ATCC-CRL-2315)), hormone receptor-positive (ER^+^, PR ^+^, HER-2) breast cancer cell line (MCF-7) (ATCC: HTB-22), and non-tumorigenic breast epithelial cell line (MCF-12A) (ATCC: CRL-10782, a kind gift from Anna-Mart Engelbrecht, Stellenbosch University, Stellenbosch, South Africa) were maintained in culture in a 9% CO_2_ incubator at 37 °C. The HCC70 cell line was cultured in RPMI-1640 media (Gibco, Trenton, MI, USA) supplemented with 10% (*v*/*v*) fetal bovine serum (FBS) (BioWest, Nuaillé, France), 1% (*v*/*v*) 100 U/mL penicillin, streptomycin, and amphotericin (PSA) (Lonza, Basel, Switzerland), 1% (*v*/*v*) GlutaMAX^TM^ (Gibco) and 0.25% (*v*/*v*) sodium bicarbonate (Gibco). The MCF-7 cell line cell was cultured in Dulbecco’s Modified Eagles Medium (DMEM) (Gibco) supplemented with 10% (*v*/*v*) FBS, 1% (*v*/*v*) PSA and 1% (*v*/*v*) GlutaMAX^TM^. The MCF-12A cell line was cultured in DMEM/Ham’s F-12 (BioWest) supplemented with 10% (*v*/*v*) FBS, 1% (*v*/*v*) PSA, 20 ng/mL human epidermal growth factor (Sigma, St. Louis, MO, USA), 100 ng/mL cholera toxin (Sigma), 500 ng/mL hydrocortisone (Abcam, Cambridge, UK) and 10 μg/mL insulin (Novo Nordisk, Bagsværd, Denmark). All cells were seeded into 96-well plates at a density of 5000 cells per well and allowed to adhere for the night in a 9% CO_2_ incubator at 37 °C. The cells were subsequently treated for 96 h at 37 °C in an incubator with 9 percent CO_2_ with the isolated compounds with six 5-fold dilutions at concentrations ranging from 15.63 to 500.00 μM or with 0.2% (*v*/*v*) DMSO (Sigma) as a vehicle control. Thereafter, 0.54 nM of resazurin (Glentham Life Sciences, Corsham, UK) solution was added, and cells were then cultured for two to four hours at 37 °C in an incubator with 9% CO_2_. A Spectramax spectrophotometer (Molecular Devices, Sunnyvale, CA, USA) was used to measure the fluorescence of the resulting solutions. The excitation and emission wavelengths were set at 560 nM and 590 nM, respectively. The experiment was repeated in technical triplicate, and the data were analyzed using GraphPad Prism (Boston, MA, USA). Half-maximal inhibitory doses (IC_50_ values) were determined using non-linear regression with Etoposide serving as a positive control. The selectivity index (SI) value of each compound was calculated as the ratio of IC_50_ of MCF-12A/IC_50_ of HCC70 or MFC-7 of the same compound, where an SI > 1 implies toxicity that is preferentially toxic to cancer cells compared to non-cancerous cells.

## 4. Conclusions

Eight compounds were isolated from the stem bark of *P. angolensis*, including Friedelan-3-one (**1**), 3α-hydroxyfriedel-2-one (**2**), 3-Hydroxyfriedel-3-en-2one (**3**), (3β)-lup-20(29)-en-3-ol (**4**), stigmasta-5-22-dien-3-ol (**5**), be 4-*O*-methyl angolensin (**6**), (3β)-3-Acetoxyolean-12-en-28-oic acid (**7**), and as tetradecyl (E)-ferulate (**8**). Compounds **1**, **2**, **3**, **4**, **5**, **6**, and **8** were isolated from the DCM crude extract of cultivated *P. angolensis*, while five compounds, **1**, **2**, **4**, **5**, and **6**, were obtained from the wild crude DCM extract. Compound **7** was isolated from the ethyl acetate extract of wild and cultivated crude extracts. The DCM extracts obtained from the stem bark of cultivated *P. angolensis* yielded more isolated compounds (**1**, **2**, **3**, **4**, **5**, **6**, and **8**) compared to the wild ones (**1**, **2**, **4**, **5**, and **6**); this could be attributed to the variation in the growing conditions. For the first time, compounds **2**, **3**, **7**, and **8** were isolated from the stem bark of *P. angolensis*. This represents the first report of the cytotoxicity of isolated compounds **1**, **2**, **3**, **6**, and **8** against cancer cells. Overall, the results show that *P. angolensis* is a rich source of bioactive constituents that can be used as anticancer agents or act as lead compounds in finding anticancer drugs.

## Figures and Tables

**Figure 1 plants-13-00301-f001:**
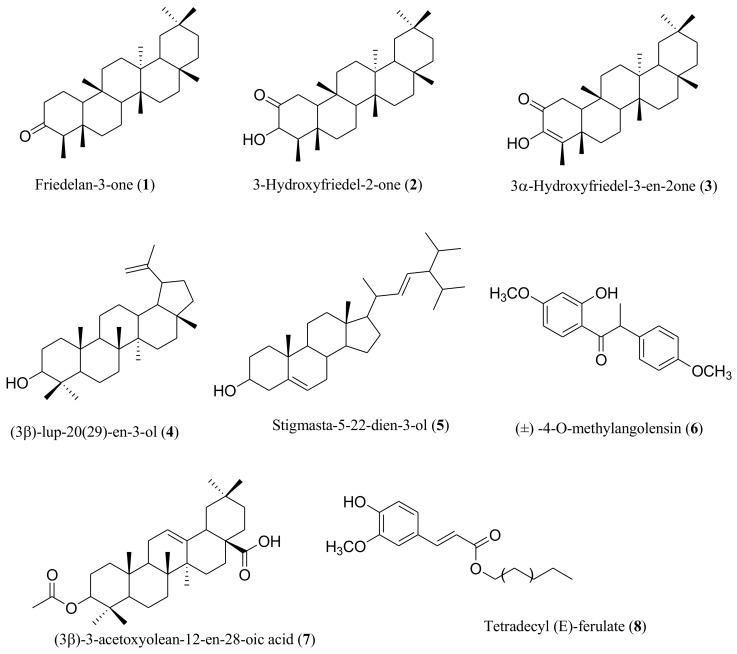
Chemical compositions of isolated compounds from *P. angolensis*.

**Table 1 plants-13-00301-t001:** Physiochemical properties of the isolated compounds.

Compound Name	[*α*]_D_ ^19.9°^	MP (°C)	IR (cm^−1^)	[M + H]^+^
*Friedelan-3-one* (**1**)	−42.75	261–263	2928 (C-H), 1715 (C=O)	427.3805
*3α-Hydroxyfriedel-2-one* (**2**)	−31.65	248–251	3444 (O-H), 2939 (C-H), 1662.6 (C=O)	443.3447
*3-Hydroxyfriedel-3-en-2-one* (**3**)	−31.65	265–268	3377.5 (O-H), 2942.7 (C-H), 1709 (C=O), 1464 (C=C), 1385.6 (C-O)	441.2928
*(3β)-lup-20(29)-en-3-ol* (**4**)	+42.44	123–125	3314.4 (O-H), 2949.7 (C-H), 1605 (C=O)	427.3677
*Stigmasta-5-22-dien-3-ol* (**5**)	+42.44	169–170	3416.3 (OH), 2939 (C-H), 1683.4 (C=C), 1045.8 (C-O)	413.3239
(±)-4-*O*-*Methylangolensin* (**6**)	−62.71	70–71	3455 (O-H), 1608 (C=O) 1451 (C=C) 1230 (C-O)	287.1341
(3β)-3*-Acetoxyolean-12-en-28-oic acid* (**7**)	+57.49	119–122	2934 (C-H), 1732, 1697 (C=O), 1462 (C=C), 1246 (C-O)	499.3795
*Tetradecyl (E)-ferulate* (**8**)	+1.06	68–70	3381 (OH), 1709 (C=O) at 1635	391.2914

**Table 2 plants-13-00301-t002:** Cytotoxicity (IC_50_) of isolated compounds from *P. angolensis* against HCC70, MCF-1 and MCF-12A.

Compound	IC_50_ (µM) HCC70	R^2^	IC_50_ (µM) MCF7	R^2^	IC_50_ (µM) MCF12A	R^2^
Friedelan-3-one (**1**)	Non-Toxic		Non-Toxic		43.86	0.52
3α-Hydroxyfriedel-2-one (**2**)	Non-Toxic		157.0 SI = 0.53	0.91	84.20	0.76
3-Hydroxyfriedel-3en-2-one (**3**)	146.4 SI = 0.93	0.98	100.8 SI = 1.35	0.97	136.0	0.99
(3β)-lup-20(29)-en-3-ol (**4**)	Non-Toxic		148.7 SI = 0.25	0.61	36.60	0.65
Stigmasta-5-22-dien-3-ol (**5**)	109.4 SI = 0.40	0.80	226.9 SI = 0.19	0.86	44.14	0.49
(±)-4-*O*-methylangolensin (**6**)	111.4 SI = 1.32	0.98	77.46 SI = 1.91	0.97	147.9	0.98
(3β)-3-Acetoxyolean-12-en-28-oic acid (**7**)	146.8 SI = 0.97	0.98	83.06 SI = 1.72	0.99	143.0	0.99
Tetradecyl (E)-ferulate (**8**)	128.2 SI = 0.05	0.56	223.1 SI = 0.03	0.86	6.95	0.54
Etoposide	44.83 SI = 0.24	0.93	15.38 SI = 0.71	0.91	10.94	0.97

SI: Selectivity Index.

## Data Availability

Data are contained within the article.

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
