# Peer review of "Phytochemicals from Pterocarpus angolensis DC and Their Cytotoxic Activities against Breast Cancer Cells"

_plants, 2024, doi:10.3390/plants13020301_

Round 1

Reviewer 1 Report

Comments and Suggestions for Authors

Comments on the manuscript titled “Phytochemicals from Pterocarpus angolensis and their cytotoxic activities against breast cancer cells”:

The authors note that, to their knowledge, no anticancer activities have been reported for the extracts and isolated compounds of P. angolensis. Furthermore, there is no available information on the geographical variation of metabolites within P. angolensis ecotypes. Their study primarily focuses on evaluating the cytotoxic properties of the crude extract and its phytochemicals. In essence, the study revolves around the isolation of phytochemicals and their subsequent assessment for biological effects. Overall, the compounds demonstrated either no toxicity or very low toxicity across all three tested cell lines. This implies that further investigation is necessary to elucidate the biological effects of the isolated substances.

Major points:

1.            The compound formulas in Figure 1 should be improved, as letters and numbers seem to merge together.

2.            The Result subsection is relatively concise; therefore, the physicochemical properties of compounds could be better presented in a table, reducing the overcrowding of text with numerical values.

3.            If climate conditions are mentioned as possible explanations for the different chemical compositions of the extracts, providing temperature values for the Limpopo Province (where the wild populations were collected) in comparison to the Gauteng Province would help clarify the extent of the temperature difference.

4.            Table 1, presenting the Cytotoxicity (IC50) of isolated compounds from P. angolensis against HCC70, MCF-1, and MCF-12A, should be relocated to the Results subsection instead of being placed in the Discussion.

5.            It is unclear why etoposide is not used as a reference for MCF-1 cells, and why no positive control is included at all.

6.            In lines 372-373, the concentrations of isolated compounds are presented in molar units that seem excessively high. It is suggested to use micromolar or millimolar concentrations. Additionally, clarity could be improved by specifying the number of concentrations tested and the dilutions between 15.63 and 500 M.

7.            The process of cell culturing is not described, including details about the media, supplements, antibiotic mix, etc. Providing this information would enhance the transparency and reproducibility of the study.

Reviewer 2 Report

Comments and Suggestions for Authors

Dear Editor,

After carefully reviewing the manuscript ‘‘Phytochemicals from Pterocarpus angolensis and their cytotoxic 2 activities against breast cancer cells’’ I find it suitable for publication after minor changes given bellow.

1. Figure 1 is of poor quality, and inadequate and robust presentation of structures, please correct it.

2. Results section regarding the cytotoxicity studies is missing and given with the discussion. Please mend it this and make it uniform.

3. Some discussion why some compounds are more active than others in order should be added.

4. Conclusions should highlight what is the applicability of the results.

Reviewer 3 Report

Comments and Suggestions for Authors

Phytochemicals from Pterocarpus angolensis and their cytotoxic activities against breast cancer cells

Zecarias W Teclegeorgish et al

Abstract

Pterocarpus anglonesis an indigenous medicinal plant belonging to the Pterocarpus genus of the Fabaceae family. It is used to treat, stomach problems, headaches, mouth ulcers, malaria, blackwater fever, gonorrhea, ringworm, diarrhea, heavy menstruation, and breast milk stimulation. Column chromatography of the stem bark extracts resulted in the isolation of eight compounds which included; friedelan-3-one (1), 3α-hydroxyfriedel-2-one (2), 3-hydroxyfriedel-3-en-2-one (3), lup-20(29)-en-3-ol (4), Stigmasta-5-22-dien-3-ol (5), 4-O-methylangolensis (6), (3β)-3-acetoxyolean-12-en-28-oic acid (7) and tetradecyl (E)-ferulate (8). The structures were established based on NMR, IR, and MS spectroscopic analysis. Triple-negative breast cancer (HCC70), hormone receptor-positive breast cancer (MCF-7), and non-cancerous mammary epithelial cell lines (MCF-12A) were used to test the compounds' cytotoxicity. Overall, the compounds showed either no toxicity or very low toxicity to all three cell lines tested, except for the moderate toxicity displayed by lupeol (4) towards the non-cancerous MCF-12A cells, with an IC50 value of 36.60 μM. Compound 3β)-3-acetoxyolean-12-en-28-oic acid (7) was more toxic towards hormone-responsive (MCF-7) breast cancer cells than either triple-negative breast cancer (HCC70) or non-cancerous breast epithelial (MCF-12A) cells (IC50 values of 83.06 vs 146.80 and 143.00 μM, respectively).

Comment to authors:

The authors provided eight compounds isolated from the stem bark extracts of Pterocarpus angolensis.

The cytotoxicity activities against breast cancer cells were investigated. However, this manuscript contains many errors. The figures were not prepared with good resolution, and the relative configuration was not carefully handled. The results are very common and do not provide any new compounds. Therefore, I regret to inform you that this manuscript may not attract many readers.

Comments on the Quality of English Language

Moderate editing of English language required

Reviewer 4 Report

Comments and Suggestions for Authors

        After reviewing the following manuscript entitled "Phytochemicals from Pterocarpus angolensis and their cytotoxic 2 activities against breast cancer cells (Plants - 2784620), I sent the following comments and observations that the authors should attend to before its publication in this journal.

I appreciate the work of the authors, but please resolve the following data: Binomial name is not written correctly (the initials of the person who gave the name should be mentioned).

The origins of the reagents and cell lines used in the study should be mentioned.

Round 2

Reviewer 1 Report

Comments and Suggestions for Authors

The manuscript is significantly improved. I do not have comments.

Reviewer 3 Report

Comments and Suggestions for Authors

The authors have made significant revisions to the revised manuscript. I am satisfied with this revision; therefore, I recommend accepting this current form.

Comments on the Quality of English Language

Minor editing of English language required